# Abiotic formation of condensed carbonaceous matter in the hydrating oceanic crust

Marie Catherine Sforna[1,5], Daniele Brunelli [1,2], Céline Pisapia [3,4], Valerio Pasini[1,3], Daniele Malferrari[1] & Bénédicte Ménez [3]

Thermodynamic modeling has recently suggested that condensed carbonaceous matter should be the dominant product of abiotic organic synthesis during serpentinization, although it has not yet been described in natural serpentinites. Here we report evidence for three distinct types of abiotic condensed carbonaceous matter in paragenetic equilibrium with low-temperature mineralogical assemblages hosted by magma-impregnated, mantle-derived, serpentinites of the Ligurian Tethyan ophiolite. The first type coats hydroandraditic garnets in bastitized pyroxenes and bears mainly aliphatic chains. The second type forms small aggregates (~2 μm) associated with the alteration rims of spinel and plagioclase. The third type appears as large aggregates (~100–200 μm), bearing aromatic carbon and short aliphatic chains associated with saponite and hematite assemblage after plagioclase. These assemblages result from successive alteration at decreasing temperature and increasing oxygen fugacity. They affect a hybrid mafic-ultramafic paragenesis commonly occurring in the lower oceanic crust, pointing to ubiquity of the highlighted process during serpentinization.

[1] Dipartimento di Scienze Chimiche e Geologiche, Università di Modena e Reggio Emilia, Via Campi, 103, 41125 Modena, Italy. [2] Istituto di Scienze del Mare–ISMAR-CNR, Via Gobetti, 101, 49100 Bologna, Italy. [3] Institut de Physique du Globe de Paris, Sorbonne Paris Cité, Université Paris Diderot, CNRS, UMR 7154. 1 rue Jussieu, 75238 Paris cedex 5, France. [4] Synchrotron SOLEIL, Campus Paris-Saclay, L'orme des merisiers, Saint-Aubin BP48, 91192 Gif-sur-Yvette cedex, France. [5] Present address: University of Liège, Department of Geology, Allée du six Août, 14, Sart Tilman, B-4000 Liège, Belgium. Correspondence and requests for materials should be addressed to M.C.S. (email: marie.sforna@unimore.it)

At slow and ultraslow Mid-Ocean Ridges (MORs), mantle-derived rocks are progressively serpentinized by aqueous fluids that circulate through the upper lithosphere. By producing $H_2$-bearing fluids, these environments are considered to be favorable for the chemical reduction of magmatic inorganic carbon species ($CO/CO_2$) or seawater carbonate ions[1,2]. Up to now, abiotic methane ($CH_4$), short-chain hydrocarbons and carboxylic acids were accordingly identified in hydrothermal fluids discharged at MORs[3–7] or in products from analog experiments[1,8,9]. However, experimental studies and thermodynamic calculations have recently shown that carbonaceous phases should be dominantly produced during serpentinization[10,11]. Yet natural occurrences of Condensed Carbonaceous Matter (CCM) were rarely reported. Graphitic material has been described in association with ultramafic rocks from different ophiolites, e.g., refs. [12,13] while poorly-structured CCM aggregates, sometimes considered as biologic in origin, have been identified in serpentinites from the Mid-Atlantic Ridge[14–16] and in serpentinized gabbroic and peridotitic xenoliths[17–19].

The samples here described were collected in the Northern Apennine ophiolites, which are lithospheric remnants of the Piedmont-Ligurian oceanic basin, a branch of the Mesozoic Tethys[20] (Fig. 1). The oceanic sequence is formed by mantle peridotites, once exposed at the seafloor, intruded by sparse gabbroic bodies and a discontinuous basaltic cover. This association presents strong similarities with present-day non-volcanic passive margins and ultra-slow spreading ridge settings[20,21]. Our study focuses on a small serpentinitic body (150 m-long; Fig. 1b) pertaining to the Val Baganza unit of the External Ligurides[22].

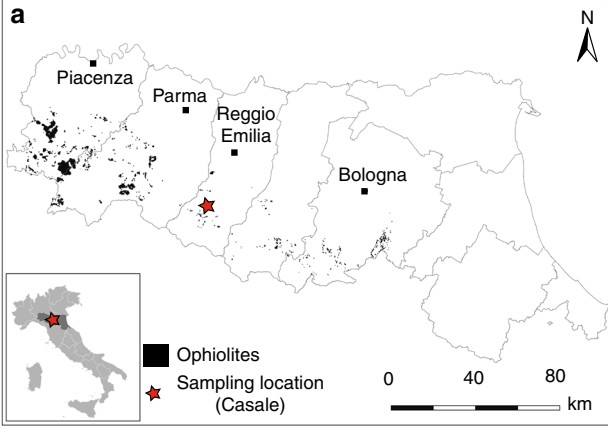

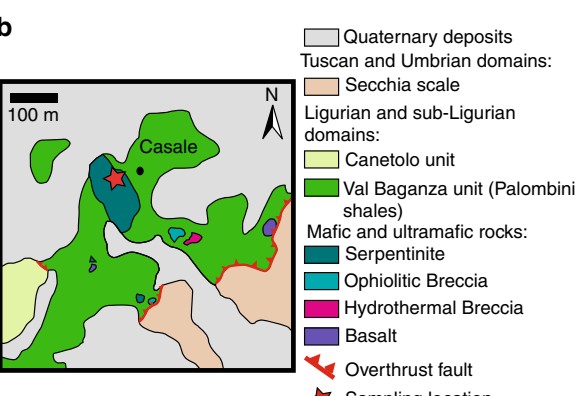

**Fig. 1** Sampling location and geological maps. **a** General map of the ophiolite outcrops in Emilia-Romagna (Italy) and sampling location (red star) modified from ref. [54]. **b** Geological map of the sampling location, modified from ref. [55]

Mineral paragenesis and nature of the organic matter disseminated in these serpentinites confirm previous studies showing that this unit was not affected by metamorphic overprint during orogenetic exhumation and obduction[23].

## Results

**A multistage aqueous alteration history.** The protolith of the studied serpentinite is a mantle-harzburgite, equilibrated in the spinel-stability field, impregnated at high temperature (~1200 °C) by a percolating melt as attested by the presence of secondary magmatic spinel and plagioclase ghosts (Supplementary Fig. 1). The progressive retrograde hydration trend of the protolith is recorded by a sequence of temperature-decreasing parageneses. The pervasive high-temperature serpentinization assemblage ($T > 300\,°C$)[24] is characterized by mesh textured lizardite + magnetite and bastite (i.e., fine-grained lizardite), substituting primary olivine and orthopyroxene, respectively (Supplementary Fig. 1, Supplementary Tables 1, 2). These textural and mineralogical relationships closely match those formed during present-day serpentinization of abyssal mantle peridotites[24].

The serpentine groundmass presents discrete microtextural domains hosting different parageneses. Residual and magmatic spinels developed inward ferritchromite (Ftc) rims (Supplementary Fig. 1, Supplementary Tables 3, 4). The ferritchromite is itself composed of a submicrometric association of Cr-magnetite (Cr-Mag), chlorite and lizardite; it displays low $Fe^{3+}\#$ ($100 \times Fe^{3+}/(Fe^{3+} + Cr + Al)$; mean = $25.87 \pm 12.76$; Supplementary Table 4), suggesting low temperatures of alteration ($\leq 200\,°C$)[25]. Ferritchromite is commonly surrounded by outward chlorite rims (Chl1), resulting from the interaction between the surrounding serpentine and the cations ($Mg^{2+}$, $Al^{3+}$) released during the spinel alteration[26] (Fig. 2, Supplementary Fig. 1, and Supplementary Tables 5, 6).

After the development of the ferritchromite rims we observe four low-temperature micro-parageneses. The first, named BastHadr, is composed of elongated and chemically-zoned hydroandraditic garnets (Hadr-1; Fig. 3) hosted in bastitic serpentine replacing orthopyroxene. The hydroandradites are rich in Ti and Cr (Supplementary Table 7) and grew following the clinopyroxene exsolution lamellae within orthopyroxene. They display narrow and elongated cavities filled with lizardite (Fig. 3c). The second paragenesis is composed of cryptocrystalline clino-chlore ± lizardite ± hydroandradite pseudomorphs on plagioclase (Chl2 ± Srp ± Hadr-2; Fig. 2, Supplementary Tables 8, 9). Unlike Hadr-1, these hydroandradites (Hadr-2) are made of nanocrystal selvages, poor in Cr and Ti (Supplementary Table 9). The third assemblage, made of Fe-rich Mg-saponite and hematite (HemSap; Fig. 2) is located only within the plagioclase ghosts, suggesting that it derives directly from plagioclase relicts or from the Chl2 ± Srp ± Hadr-2 assemblage. Although a replacement of plagioclase relicts cannot be completely excluded, the HemSap domains and the Chl2 ± Srp ± Hadr-2 domains clearly intertwine (Fig. 2b, c), with a decrease of the chlorite and hydroandradite proportions toward the domain borders, in favor of a progressive inward alteration of the Chl2 ± Srp ± Hadr-2 domain to saponite. Local interlayering of chlorite within saponite supports local replacement of Chl2 ± Srp ± Hadr-2 pseudomorphs after plagioclase (Supplementary Tables 10, 11). Saponite displays high content in Fe (up to ~7%Wt; Supplementary Table 10). In the same domains, hematite (Supplementary Table 12) forms rosettes interlocked with serpentine, saponite and chlorite. Hematite is always found in association with saponite but the reverse is not true, suggesting that hematite grew at the expense of saponite. Variably altered spinels are commonly preserved within the pseudomorphs (Fig. 2). The fourth low-temperature paragenesis

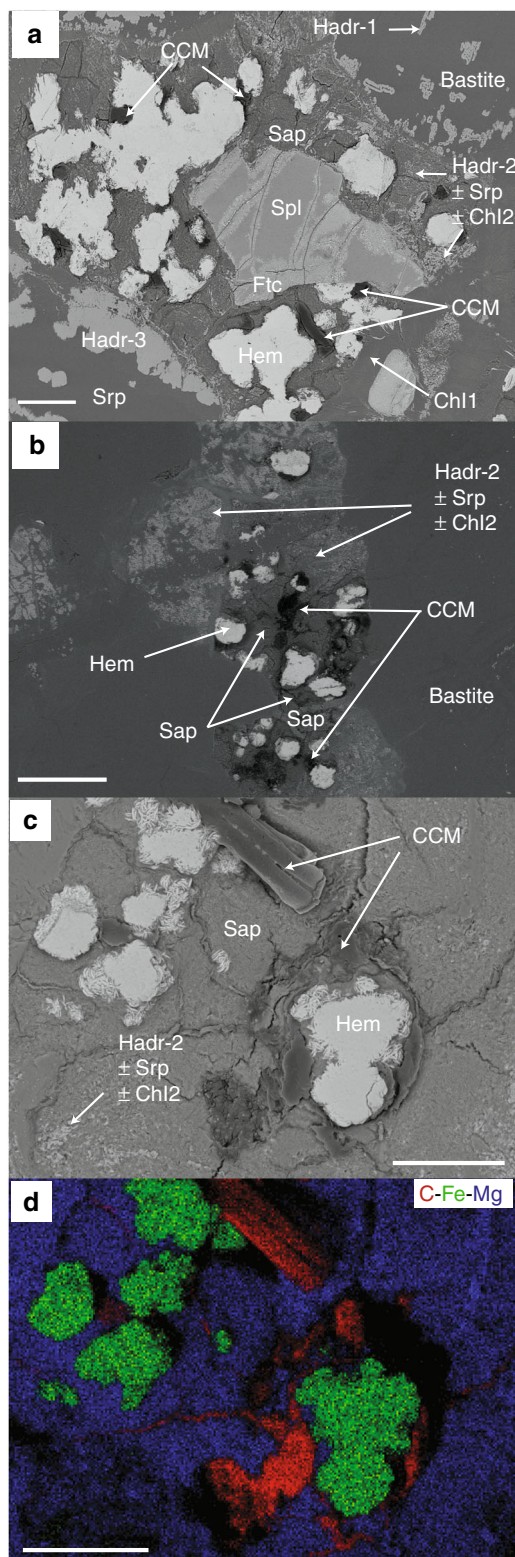

**Fig. 2** Association of large patches of Condensed Carbonaceous Matter (CCM) with hematite (Hem) and saponite (Sap) assemblage (HemSap) shown by backscattered electron SEM images . **a** The HemSap assemblage grew at the expense of chlorite (Chl2) ± serpentine (Srp) ± type 2-hydroandradite (Hadr-2) resulting from plagioclase pseudomorphosis. The former plagioclase surrounded primary spinel (Spl) now displaying limited ferritchromite (Ftc) rims. **b** Same assemblage grew over the pseudomorphs of former interstitial plagioclase. **c** Large patches of CCM coating hematite and invading microcracks affecting saponite. **d** Associated elemental distributions of carbon (red), iron (green) and magnesium (blue). Scale bars correspond to 200 μm

electron microscopy (SEM) and characterized by Raman and Fourier transform-infrared (FTIR) spectroscopies. CCM occurrences are associated with three out of the four low-temperature micro-parageneses described above, namely the bastite-hosted hydroandradite Hadr-1 (CCM-BastHadr), the pseudomorphosed plagioclase (Chl2 ± Srp ± Hadr-2) coupled to the ferritchromite rims (CCM-PPFtc), and the hematite + Fe-rich Mg-saponite assemblage (CCM-HemSap) (Figs. 2–5). CCM aggregates display differences in size and composition among the various mineralogical assemblages. The rare and very small (~2 μm) CCM-PPFtc occurrences (Fig. 4) were not analyzed by Raman and FTIR spectroscopy due to intense fluorescence and their too small size, respectively.

The CCM-BastHadr appears as thin films (< 6 μm) coating Hadr-1 crystals and their inner cavities (Fig. 3). The associated Raman spectrum (Fig. 5, Supplementary Table 15) shows two large bands in the 1100-1450 cm$^{-1}$ region, corresponding to $Cr^{3+}$ luminescence in garnet[27]. Although these bands partially mask the organic signal, $CH_2$ and $CH_3$ vibration bands can still be recognized on their shoulders. Intense bands corresponding to aliphatic COOH (1532 cm$^{-1}$)[28] can be observed in the 1400-1800 cm$^{-1}$ region. Another band corresponding to $COO^-$ stretching can be detected but no clear C=C vibrations characteristic of aromatic moieties were identified[29]. The CCM-HemSap assemblages form large aggregates up to 100–150 μm in size. They are mainly associated with hematite, filling the mineral embayments and cracks developed during the retrograde transformation of Chl2 ± Srp ± Hadr-2 into Fe-rich Mg-saponite (Fig. 2). The associated Raman spectrum displays $CH_3$ and $CH_2$ vibrations suggesting the presence of aliphatic chains, as confirmed by FTIR spectroscopy (Supplementary Fig. 2). The mean methylene to methyl ratio $R_{CH2/CH3}$ (1.47 ± 0.45; Supplementary Table 16) indicates short aliphatic chains, bearing up to 6–8 carbon atoms[30]. Contrarily to CCM-BastHadr, CCM-HemSap assemblages display several vibration bands characteristic of aromatic moieties (Fig. 5, Supplementary Table 15).

## Discussion

The strict spatial association between the organic and mineralogical phases suggests that the different forms of CCM are in paragenetic equilibrium with secondary mineral phases. This observation, supported by the systematic association of a given CCM type with a given mineral paragenesis, indicates that CCM formation could have occurred simultaneously to the growth of the host mineralogical assemblage, thus suggesting an abiotic endogenic genesis. Additionally, the CCM spectral signatures reported here (Fig. 5) lack evidence of protein-forming amide groups of biological origin, such as those previously reported in oceanic serpentinites[14,15]. A CCM genesis after thermal degradation of pristine biogenic material can also be excluded based on Raman spectra that do not display the expected broad bands of

contains large hydroandradites (Hadr-3; Fig. 2a and Supplementary Table 13) developed on the edges of serpentine veins (Supplementary Table 14), growing along serpentine fibers. Their $Cr_2O_3$ and $TiO_2$ contents are similar to Hadr-2 and Hadr-1, respectively (Supplementary Table 13).

**Multiple occurrences of condensed carbonaceous matter.** Significant amounts of CCM have been identified by scanning

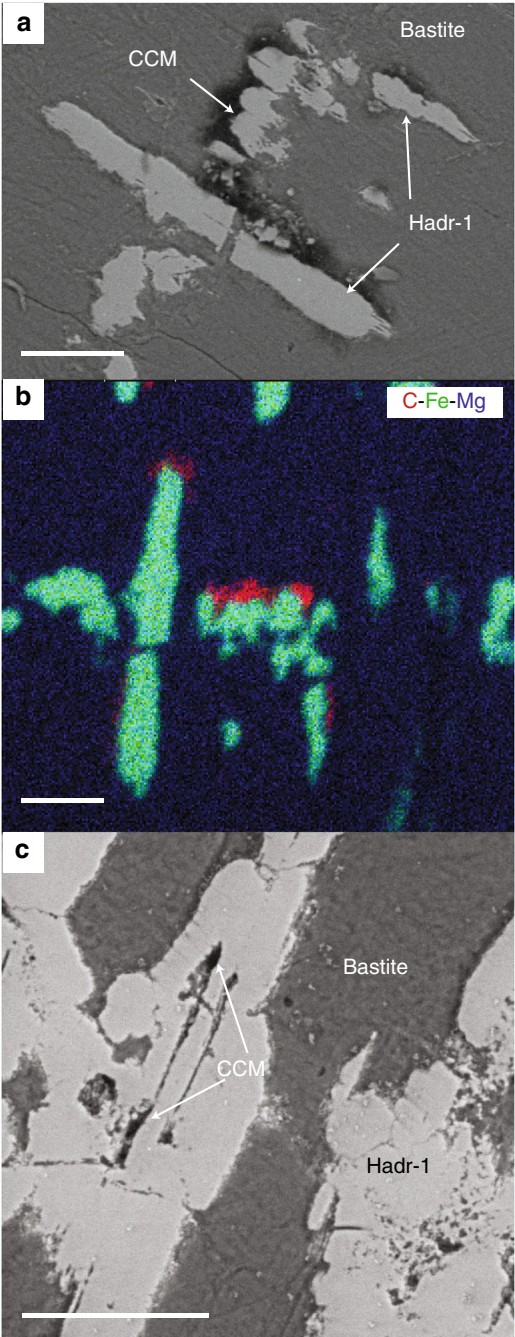

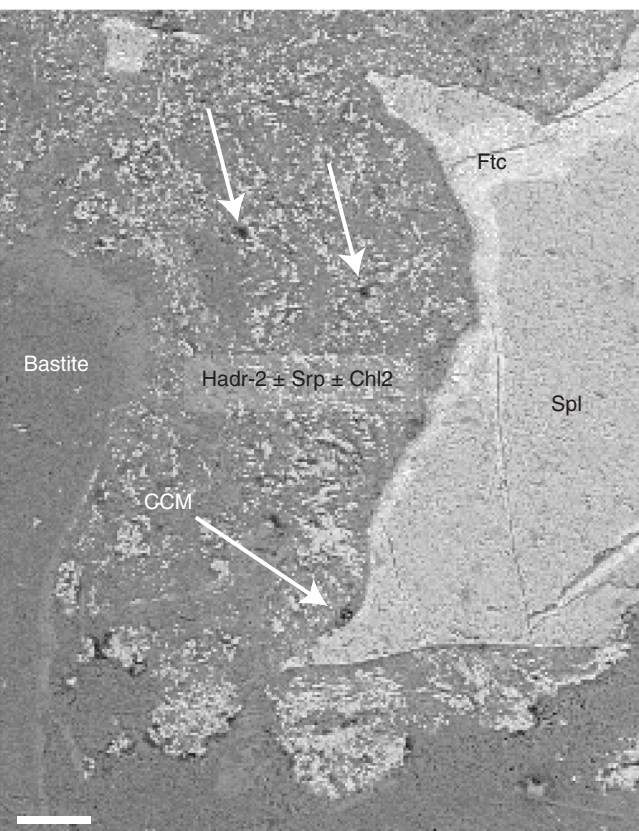

**Fig. 4** Association of CCM with ferritchromite (Ftc) rims contouring altered primary spinel and pseudomorphosed plagioclase substituted by Chl2 ± Srp ± Hadr-2 (PPFtc assemblage). Backscattered electron SEM images showing the presence of rare and small patches of condensed carbonaceous material (CCM-PPFtc), highlighted by white arrows. Scale bar corresponds to 50 μm

**Fig. 3** Association of CCM with type 1-hydroandradite (Hadr-1) and bastite assemblage (BastHadr). **a** Backscattered electron SEM image showing a thin organic film coating the surface of Hadr-1 growing subparallel to the exsolution planes of the primary pyroxene. **b** Associated elemental distributions of carbon (red), iron (green) and magnesium (blue). **c** Backscattered electron SEM image showing that CCM is also present in the cavities of Hadr-1. Scale bars correspond to 20 μm

graphitic carbon (~1340–1360 and 1580–1610 cm$^{-1}$)[31] that would be produced by this process.

Casale samples show that the amount of CCM varies among the observed mineral assemblages: minor in the plagioclase pseudomorphs coupled to the ferritchromite rims (CCM-PPFtc), more important in the BastHadr domains (CCM-BastHadr) and massive in the HemSap paragenesis (CCM-HemSap). The last two CCM pools while presenting a similar aliphatic character,

differ in their carboxylic pattern: the CCM-BastHadr assemblage shows a higher content in carboxylic functional groups whereas aromatic carbon is present in the CCM-HemSap assemblage. While the hydration of olivine was considered as the only putative reaction pathway for CCM formation in recent thermodynamic calculations[11], these observations are suggestive of differential formation pathways and productivity possibly implying differences in the synthesis mechanism and limiting factors at local scale.

The formation of non-graphitic CCM in serpentinization-related systems was thermodynamically predicted to occur during olivine serpentinization in the 200–400 °C temperature interval[11]. At these temperatures CCM forms preferentially provided that $CH_4$ is kinetically inhibited[2,8,9,32–34]. Our data confirm and significantly extend CCM abiotic synthesis to diverse mafic and ultramafic mineral precursors and lower temperature ranges. Overall, the low-temperature parageneses found in the Casale serpentinite are progressive steps of a low-temperature alteration sequence resulting from increasing oxidizing conditions occurring during the uplifting of the oceanic mantle and its exposition to the seafloor. The crystallization and stability fields of these low-temperature assemblages give constraints on the formation of CCM-BastHadr and CCM-PPFtc assemblages during hydrothermal alteration at depth in an oceanic subaxial environment (< 10 km). The crystallization of hydroandradite occurs at temperatures < 200 °C, low oxygen fugacity, low $CO_2$ partial pressure and low silica activity[35]. The CCM-HemSap assemblage was formed later at lower temperatures, higher silica activity and in a

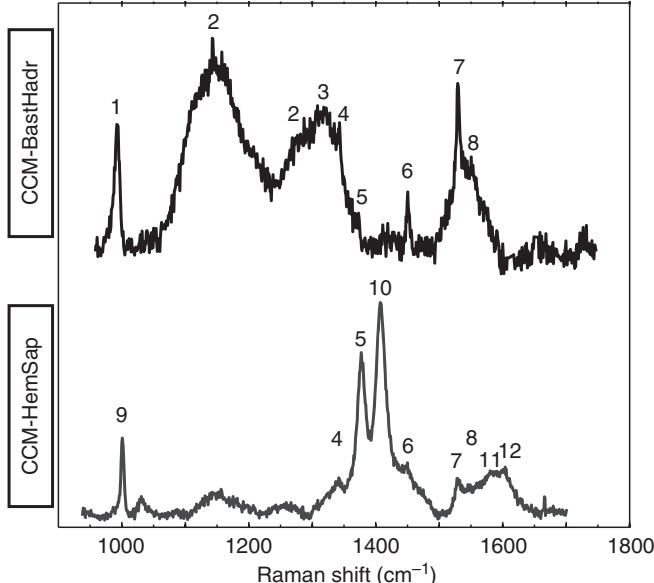

**Fig. 5** Raman spectra for CCM-BastHadr and CCM-HemSap assemblages and assignment of the different bands. 1: Hydroandradite, 2: $Cr^{3+}$ luminescence, 3: $CH_2$ wagging, 4: CH deformation, 5: $CH_3$ in aliphatic compounds ($CH_3$ symmetric deformation), 6: $CH_2$ and $CH_3$ in aliphatic chains (bending, scissoring, antisymmetric stretching), 7: Aliphatic COOH, 8: $COO^-$ antisymmetric stretching, 9: C-C aromatic or phenyl, 10: $COO^-$ symmetric stretching, 11: C=C aromatic, 12: C-C aromatic ring stretching. Assignment was done using refs. [28, 29]. For the exact position of the bands see Supplementary Table 15

slightly-more oxidized environment, as suggested by the presence of Fe-rich Mg-saponite and hematite[35,36]. Saponite, a common secondary or tertiary product of hydrothermal alteration of mafic and ultramafic minerals[35-39] can form under a wide range of temperatures, from 20 to 300 °C[40,41]. The saponite enrichment in Fe and its mixed valence can be related to slow alteration at low temperature (<150 °C) in the upper most portion of the crust during uplifting and progressive exposure at the seafloor of mantle-derived rocks to cold and oxidizing seawater-derived fluids at increasing water/rock ratio as the reaction progresses[41-44]. Similar or close conditions are likely kept during the crystallization of hematite linked to the destabilization of Fe-rich Mg-saponite by oxidizing fluids[36]. When saponite is oxidized, nontronite is generally the main by-product at high water/rock ratios (ca. 1000)[36]. On the contrary, at low water/rock ratios (ca. 1), hematite becomes an additional by-product of saponite oxidation. It can be the sole product if the $O_2$ dissolved in the fluids is as low as 0.008–0.010 g $O_2$ per gram of rock[36].

In low-temperature serpentinizing environments, the reduction of inorganic carbon to organic compounds is partly driven by the availability of $H_2$. Its generation results from water reduction associated with oxidation of $Fe^{2+}$-bearing minerals (olivine, pyroxene and serpentine)[45,46]. Above 300 °C, this reaction is fast and efficient, and a large quantity of $H_2$ is produced while the resulting $Fe^{3+}$ is stored mainly in magnetite[45-47]. Below 200 °C, the production of $H_2$ and hence the abiotic synthesis potential should decrease because of slower reaction kinetics and partitioning of $Fe^{2+}$ into secondary minerals[45-48]. Nonetheless, the low-temperature formation of ferric-serpentine and hydroandradite as serpentinization reaction by-products has been suggested as an efficient process for $H_2$ production[49,50]. Similarly, the formation of ferritchromite rims where spinel $Fe^{2+}$ is oxidized in $Fe^{3+}$-bearing Cr-magnetite and the later crystallization of hematite are also potential sources for $H_2$[48,50].

Dihydrogen availability released by mineral reactions is likely not the only factor impacting local CCM productivity. Below 200 °C, the presence of a catalyst is required to initiate carbon reduction reactions[32]. For instance, provided that the fluid contained comparable inorganic carbon content, CCM is not systematically found in association with hydroandradite even though its formation produces 1 mole of $H_2$ per mole of hydroandradite crystallized[50]. Significantly, CCM appears associated with the hydrogarnets of the BastHadr associations (Hadr-1) that show Cr-rich rims containing up to 2.7%Wt of $Cr_2O_3$ (Supplementary Table 7). $Cr^{3+}$ is an efficient catalyst[33] and may have played a catalytic role at the surface of Hadr-1, promoting the peripheral formation of the CCM-BastHadr assemblage. Conversely, the CCM-PPFtc occurrences are limited to thin films on the spinel borders and tiny nuggets dispersed in the surrounding Chl2 ± Srp ± Hadr-2 pseudomorphs on plagioclase (Fig. 4). Hadr-2 has in fact low amount of $Cr^{3+}$ (mean $Cr_2O_3 \sim 0.4\%$ Wt; Supplementary Table 9), thus a likely low catalytic capability. Moreover, while spinels are thought to promote organic synthesis[33,51,52], the growth of $Fe^{3+}$-hydroxides at their surface during alteration can limit internal $Fe^{2+}$ oxidation and associated $H_2$ delivery[48]. This is supported by the low measured $Fe^{3+}$# of the spinel Ftc rims (Supplementary Table 4). The surrounding paragenesis is composed of $Fe^{2+}$-bearing phases (Chl1, Chl2, Srp, ferritchromite i.e., Chl, Srp, Cr-Mag; Supplementary Tables 4, 5, 6, 8) suggesting an overall limited oxidation of the primary phase and hence limited $H_2$ production. This mineral aggregation at the ferritchromite surface may also have acted as barrier for $Fe^{2+}$ diffusion from external sources that would have sustained a longer generation of $H_2$[48] and thus higher CCM production.

The most abundant CCM accumulations are found in the HemSap domains. We propose they originate from the interplay between the capability of hematite to produce $H_2$ during its formation[48,50] and the cation exchange capacity of the saponite structure[53]. Octahedral $Fe^{2+}$ and $Fe^{3+}$ for $Mg^{2+}$ and tetrahedral $Al^{3+}$ and $Fe^{3+}$ for $Si^{4+}$ heterovalent substitutions in the Fe-rich Mg-saponite silicate layers (Supplementary Tables 10, 11) may create charge unbalances in the octahedral and tetrahedral sheets[53]. Moreover, saponite appears to be highly heterogeneous at the micrometric scale with varying quantities of $Fe^{3+}$ and $Fe^{2+}$ in various coordination numbers (Supplementary Tables 10, 11). The charge unbalance promotes the exchange of cations in the clay mineral interlayers[53]. It also provides catalytic acid sites in the tetrahedral layers that may promote direct adsorption/intercalation, retention, and polymerization of organic compounds[53]. Chemical formulas calculated for the Fe-rich Mg-saponite also reveal a high octahedral occupancy that nicely agrees with the possible presence of transition metals (Cr, Ni, Fe, Mn, and Mg; Supplementary Tables 10, 11) in the saponite interlayers. These cations would further promote the complexation of organic compounds[53]. Coupled to the $H_2$ produced during the crystallization of hematite, this would promote further CCM formation.

Microscopy and microspectroscopy techniques allowed for the first time to document the occurrence of different types of abiotic condensed carbonaceous matter within natural serpentinites. These occurrences are strictly associated with particular low-temperature mineralogical assemblages, suggesting that the organic material is in paragenetic equilibrium with each mineral assemblage. As compiled in Fig. 6, the formation of the different assemblages occurs at decreasing temperatures and (slightly) increasing dissolved $O_2$ concentrations, showing that CCM formation does not occur in a single event during aqueous alteration of the oceanic lithosphere. The slight change in oxygen concentrations necessary to produce $Fe^{3+}$-oxides[36] is likely not sufficient to prevent the continuous formation of CCM given that the saponite and hematite surfaces provide locally reducing micro-domains[48]. The combination of enhanced $H_2$ production due to hematite crystallization coupled to

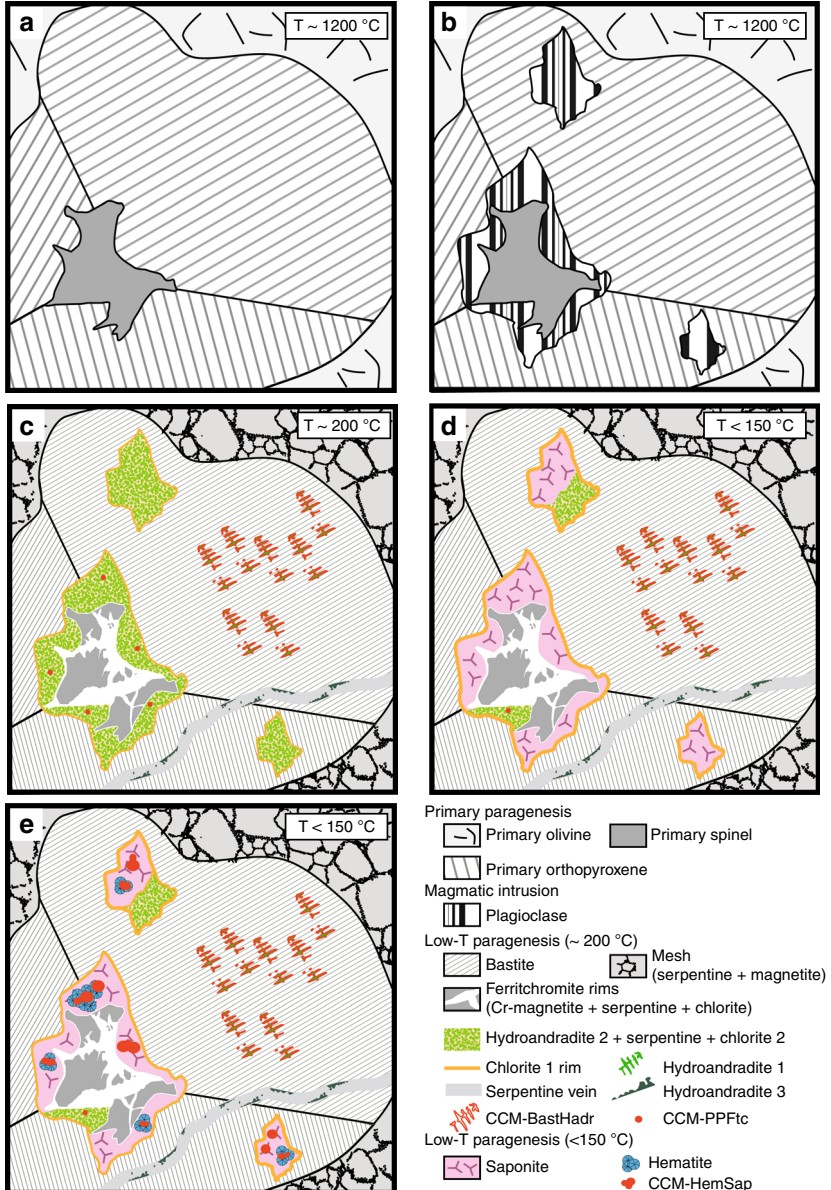

**Fig. 6** Conceptual model of the different events that lead to the generation of condensed carbonaceous matter (CCM) associated with mineralogical transitions during the low-temperature aqueous alteration of the Casale serpentinite. **a** Primary paragenesis. Mantle harzburgite equilibrated in the spinel-stability field. **b** Magmatic intrusion. Impregnation of the protolith at high temperature (T ~ 1200 °C) by percolating melts, resulting in the growth of secondary spinels and plagioclases. **c**–**e** Aqueous alteration at decreasing temperatures. **c** Serpentinization at high temperature (T > 300 °C) of the protolith, giving rise to bastite and mesh textured lizardite and magnetite, followed by low-temperature alteration (T~200 °C) of the serpentinized assemblage at low $O_2$ fugacity and low silica activity. These low-T aqueous alteration phases gave rise to (1) the growth of ferritchromite rims on spinels and the pseudomorphosis of plagioclase into cryptocrystalline clinochlore ± lizardite ± hydroandradite (Chl2 ± Srp ± Hadr-2) assemblage, (2) the growth of hydroandradite (Hadr-1) in the bastitized orthopyroxene, and (3) the growth of hydroandradite (Hadr-3) in late serpentine veins. These different parageneses served as catalyst for the formation of different types of CCM, specifically associated with Hadr-1 (CCM-BastHadr assemblage, bearing mainly aliphatic chains) and with plagioclase pseudomorphs and ferritchromite rims (CCM-PPFtc assemblage). **d** Alteration of the low-T assemblages by colder ( < 150 °C) fluids with increased silica activity and slowly-increasing $O_2$ fugacity. It resulted in the replacement of the Chl2 ± Srp ± Hadr-2 assemblage by saponite. **e** Crystallization of hematite at the expense of saponite at slightly-increased $O_2$ fugacity and low water/rock ratio. The catalytic properties of saponite and hematite gave rise to large aggregates of CCM (CCM-HemSap assemblage) formed in paragenetic equilibrium and bearing aromatic carbon and short aliphatic chains

the saponite/hematite catalytic capabilities is a possible highly efficient engine to produce large CCM accumulations. Such process could be widespread in mafic/ultramafic-hosted hydrothermal systems at MORs, thus storing effectively organic carbon below the oceanic seafloor in a relatively immobile form. The condensed carbonaceous material can be preserved on the long term like in the Casale samples here described, or serve as carbon sources for deep microbial ecosystems[14], with also the potential to impact abiotic synthesis pathways including dihydrogen or methane generation[11].

## Methods
**Sample preparation**. To limit any laboratory contamination, samples were carefully prepared in a clean organic-free environment. Inner cores of the collected samples were extracted with a saw treated with 5% sodium hypochlorite. Cutting

has been done with sterilized demineralized water. From these cores, thin sections and manually-prepared double-polished resin-free and glue-free chips were prepared using silicon carbide polishing disks.

**Scanning electron microscopy**. SEM images were collected at the Centro Inter-dipartimentale Grandi Strumenti (CIGS, Università di Modena e Reggio Emilia, Italy) on uncoated thin sections in backscattered and secondary electron modes under low vacuum using an environmental scanning electron microscope Quanta-200 (Fei Company–Oxford Instruments). Analytical conditions were 12–20 kV accelerating voltage, low current and a 10.8–11.8 mm working distance. Energy-dispersive X-ray spectroscopy analyses were carried out with an Oxford INCA-350 spectrometer. Elemental distributions and images were processed using the INCA™ software.

**Raman spectroscopy**. Raman spectra were collected on thin sections and free-standing samples (i.e., without any glue or resin) using two Jobin Yvon LabRAM™ microspectrometers (CIGS, Modena and Department of Physic and Earth Sciences of the University of Parma, Parma, Italy) using the 632.81 nm wavelength of a 20 mW He-Ne laser focused through a Olympus BX40 microscope with × 100 objective (Numerical Aperture (NA) = 0.9). Laser excitation was adjusted to an on-sample intensity inferior to 2 mW with integration time of $3 \times 100$ s, well below the critical dose of radiation that can damage carbonaceous material. This configuration yielded a planar resolution close to 1 μm. Acquisitions were obtained with a 1800 l/mm grating illuminating a Pelletier-cooled $1024 \times 256$ pixel CCD array detector. Punctual analyses were carried out in static mode with a spectral detection range of 200–2000 $cm^{-1}$. Beam centering and Raman spectra calibration were performed daily on a quartz crystal with a characteristic $SiO_2$ Raman band at 463.5 $cm^{-1}$. To exclude any organic contamination, carbonaceous matter spectra collected on conventional thin sections were systematically compared to the ones obtained on resin- and glue-free samples. Data were processed with LabRAM™ and WiRE 3.3™ softwares.

**Fourier transform-infrared microspectroscopy**. FTIR measurements were performed on a Thermo Scientific Nicolet iN10 MX imaging microscope (IPGP, Paris, France). Data were acquired with a conventional Ever-Glo™ infrared source equipped with a × 15 objective (NA = 0.7) and a liquid nitrogen cooled MCT-A detector. Spectra were collected as punctual analysis or in mapping mode. In punctual analysis mode, spectra were recorded in reflection with an incident beam collimated to a sample area of $20 \times 20$ μm², a spectral range of 600-4000 $cm^{-1}$ and a spectral resolution of 4 $cm^{-1}$. Each punctual analysis corresponded to the sum of 512 accumulations converted to absorbance with Omnic™ Picta™ software (Thermo Scientific). To increase the statistic value of the methylene to methyl ratio ($R_{CH2/CH3}$) obtained for CCM-HemSap assemblages, maps were collected in transmission on organic-rich zones. With this aim, the incident beam was collimated to a $20 \times 20$ μm² sample area, the spectral acquisition range was 600–4000 $cm^{-1}$, the spectral resolution was 8 $cm^{-1}$ and 64 spectra were accumulated for each pixel. Absorption bands in the spectral region of aliphatic moieties stretching (2800–3000 $cm^{-1}$) were fitted using a Gaussian-Lorentzian function with the WiRE 3.3™ software. $R_{CH2/CH3}$ was calculated following ref. [26] based on the intensities of the fitted asymmetric stretching bands of $CH_2$ (~2930 $cm^{-1}$; $I_{CH2}$) and $CH_3$ (~2960 $cm^{-1}$; $I_{CH3}$) ($R_{CH2/CH3} = I_{CH2}/I_{CH3}$; Supplementary Fig. 2, Supplementary Table 16).

**Electron microprobe analysis**. Mineral chemistry was characterized by electron microprobe analysis (EMPA) with the Cameca SXFive installed at CAMPARIS, University Pierre et Marie Curie (Paris, France). Thin sections were coated with carbon once all the other analyses were achieved. Operating conditions were 15 kV and ~10 nA. Analyses were acquired in punctual mode and in stage map mode (grid pattern). Maps ($300 \times 250$ pixels) were generated with a 1 μm step and a 0.1 s dwell time. For each pixel characteristic of a mineral, the percentage in weight (% Wt) of each element analyzed and mineral formula were retrieved (Supplementary Tables 1-5, 7-10, 12-14).

## Data availability

The authors declare that the data supporting the findings of this study are available within the article and its Supplementary Information File.

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

## Acknowledgements

Massimo Tonelli, Mauro Zapparoli, Fabio Bergamini (CIGS, Modena), Danilo Bersani (Università di Parma), Nicolas Rividi and Michel Fialin (CAMPARIS, Paris) are thanked for their help during data acquisition. This research was supported by PNRA (PdR 2016_00245), the Preistoria Attuale Foundation, the Deep Energy community of the Deep Carbon Observatory awarded by Alfred P. Sloan Foundation, the deepOASES ANR project (ANR-14-CE01-0008-01), the French CNRS (Mission pour l'Interdisciplinarité, Défi Origines 2018), the Italian PRIN (prot.2015C5LN35) and the Marie Curie Cofund Program at the University of Liège. This is IPGP contribution n° 3970.

## Author contributions

M.C.S., D.B., C.P., V.P., D.M., B.M. acquired and treated the data. M.C.S. D.B. and B.M. wrote the paper with input from all the co-authors.

## Additional information

**Competing interests:** The authors declare no competing interests.

