## [Peer Review File · Nature Communications]

Reviewers' comments:

Reviewer #1 (Remarks to the Author):

This is a pioneering report on the abiotic formation of condensed carbonaceous matter in mantle-derived serpentinites of ophiolites. As far as I can recall there are no (or just peripheral) publications on the occurrence of solid abiotic organic matter in mafic or ultramafic rocks on Earth. The findings will certainly have an impact on the view of the deeper - mantle - parts of the ocean floor as constituents of the global carbon cycle, although quantification may be a major challenge for the future. I have only one specific comment; it would be appropriate to include the citation to the following two articles on line 31:

Holm, N.G. and Charlou, J.L. 2001. Initial indications of abiotic formation of hydrocarbons in the Rainbow ultramafic hydrothermal system, Mid-Atlantic Ridge. *Earth and Planetary Science Letters* 191, 1-8.

Charlou, J.L., Donval, J.P., Jean-Baptiste, P., and Holm, N., 2002. Geochemistry of high H₂ and CH₄ vent fluids issuing from ultramafic rocks at the Rainbow hydrothermal field (36°14'N, MAR). *Chemical Geology* 191, 345-359.

That said, I definitely think that this contribution should be published in *Nature Communications*.

Reviewer #2 (Remarks to the Author):

This paper documents the occurrence of condensed carbonaceous material (CCM) in oceanic serpentinites, and the authors divide this CCM material into three types on the basis of mineral associations. They use Raman spectra to show that the CCM is not consistent with a biological origin, in contrast to CCM that has previously been identified in seafloor serpentinites. They suggest that minerals catalyze the production of CCM by abiogenic reactions, and infer that this process is important for the global carbon cycle. Serpentinization in general is of great current interest, from microbiologists and geochemists, and their role in cycling of volatiles and other elements is important. I think this paper is timely and worthy of publication, but I see three key questions that need to be answered before it could be ready for publication.

The finding of abiogenic CCM is new and an important part of understanding the origins of reduced carbon in oceanic serpentinites. I am not an expert in organic carbon compounds, but the evidence presented is convincing that the CCM does not have the characteristics of biogenic CCM. The authors do not, however, address the question of whether this CCM could be derived from original biogenic CCM (see comment no 1 below).

The different occurrences of CCM are linked to different low-T mineral assemblages, and this linkage is convincing. The authors do not, however, provide convincing evidence for some of their proposed mineral replacement reactions (see comment no 2 below). This may be a minor point, because the CCM is clearly associated with these different assemblages. Whether these formed by replacement of other secondary phases or replacement of igneous relics may not affect their arguments about the formation of CCM. But the authors need to better define the saponite (calculate mineral formulas, see detailed comments below), and do a better job of putting their interpretations into a realistic scenario for uplift and exposure of mantle material on the seafloor (or else do not discuss this at all; see detailed comments).

Finally, the authors overstate the importance of serpentinization on the global carbon budget. This is presented as the major significance of this work, but this question has already been addressed in several other studies. See comment no 3 below.

1. What about obduction reactions affecting CCM that originally had a biological origin in the rocks? The authors cite a geologic map for evidence that the rocks are "not affected by metamorphic overprint during orogenic exhumation and obduction." This is not very convincing. The rocks could easily have been heated to 150-200°C and the minerals not show any effects of metamorphism because they are stable at such conditions. What happens to biogenic CCM like

that observed in seafloor serpentinites if held at 200°C for thousands (tens or hundreds of thousands?) of years? Could this change biogenic CCM into the observed “abiogenic CCM” material? I do not know enough about organic matter diagenesis to know if this is possible, so this may not be significant, but this possibility needs to be addressed. Similar mantle rocks that were serpentinized on the Tethyan seafloor and are located ~80 km west of Casale were subjected to up to prehnite-pumpellyite facies conditions, but show little mineralogical effects of this because there were essentially no fluids present (only minor local effects in fractures/veins).

2. Some of the low temperature mineral reactions that are proposed are not convincing. What is the evidence for replacement reactions among low-T minerals? Specifically, what is the evidence that saponite replaces chlorite etc? What is the evidence that hematite replaces saponite? (ref 41 is misquoted about this on line 130). Such replacement reactions are not seen in oceanic rocks affected by low-T alteration. Typically highly unstable magmatic minerals are replaced by stable low-temperature phases, but low-T secondary minerals can show disequilibrium and sequential overgrowths rather than replacement (e.g., see

Alt, J.C., Very Low Grade Hydrothermal Metamorphism of Basic Igneous Rocks. In: Very Low Grade Metamorphism. M. Frey and D. Robinson, eds., Blackwell Scientific, pp. 169-201.

Alt, J.C., Alteration of the Upper Oceanic Crust: Mineralogy, Chemistry, and Processes, in: Elderfield, H. and Davis, E., eds, Hydrogeology of the Oceanic Lithosphere, Cambridge University Press, New York, 456-488.

To me it is more likely that saponite replaces relics of magmatic plagioclase, as part of sequential reactions: chlorite replaces plagioclase at >200°C, then saponite (or even mixed-layer saponite-chlorite) replaces remaining plagioclase at lower temperatures. I do not understand how hematite replaces saponite. The authors should calculate chlorite and saponite mineral formulas. This could help them constrain the conditions of their formation. The presence or absence of chlorite layers in saponite would help constrain temperatures, and the authors could test whether ferrous or ferric iron (or both) are present (relevant to their arguments about layer charge and organic molecules in interlayer positions).

3. The authors imply the significance of serpentinization for the global carbon budget, and say that this is “a potential major controller of the global cycle.” (lines 11 and 25 in the Abstract; lines 182 and 185 in the concluding section). But the amounts and significance of carbon, including organic (reduced) carbon, in seafloor serpentinites has already been documented: see Alt et al Lithos 2013, Chem Geol 2012, and Schwarzenbach et al., GCA 2012, who analyzed serpentinites from the northern Apennines and the modern seafloor for organic carbon contents and isotopic compositions. Alt et al 2013 show that only a few percent of serpentinized mantle material is present at oceanic crustal levels, and the carbon budget in serpentinites is the same as in mafic oceanic crust, so how does this make serpentinization a “major controller”? There are many (hundreds?) of analyses of carbon isotopes for organic carbon in serpentinites, how do these data fit with inorganic origins of CCM vs organic CCM? (See paper by Schwarzenbach et al 2012 GCA for data for Ligurian serpentinites).

Detailed comments:

31: analogue instead of analogical

38: ..ophiolites ,which are lithospheric

49: what do you mean by “diffuse” high-temperature assemblage? Do you mean pervasive?

50: but olivine is stable in hydrothermal systems at T>350°C. Maybe rephrase this sentence to indicate high T breakdown of pyroxene, and then olivine reaction at lower temperatures?

53: closely is better than strictly

55: rephrase: ..ferritchromite replacement rims and surrounding plagioclase was replaced by chlorite.

59: delete “contextually or”

65: (and elsewhere) pseudomorphs, not pseudomorphoses

70: commonly is better than often

71: paragenesis, not paragenese

92: associated with hematite (not to)

93: what is the evidence for replacement by saponite? Why not saponite replacing relics of plagioclase?

104: delete "in" at beginning of line; reported here, rather than here reported

105: lack evidence of protein formingderivation, such as those previously reported....

107: ...amount of CMM...

111: delete "solely"

112: ...as the only putative...

121: ...depth in an oceanic subaxial...

122: delete "in fact"

124: but Fe-saponite contains ferrous iron. It is a trioctahedral phyllosilicate, so it must be ferrous iron. You should calculate mineral formulas so you can examine the effects of varying Fe³⁺/Fe^T

128-130: this comparison is not valid for mantle material uplifted and exposed on the seafloor.

These lines describe hydrothermal conditions and processes in typical layered mafic crust and are not applicable to exposed mantle sections. See Alt et al 2013 Lithos for some indication of late, low temperature oxidative effects (seafloor weathering) in the uppermost portions of exposed mantle on the seafloor. See also papers about Ocean Drilling at the Iberian Margin for depth variations in some exposed serpentinized mantle sections.

130: ref 41 never said that hematite grows at the expense of saponite as stated here.

161: I can understand that saponite could have ferric iron in some layers, but then it is not strictly saponite but rather a dioctahedral smectite component (nontronite layers). You should calculate mineral formulas for the saponite analyses, then you can perhaps say something about ferric vs ferrous iron contents, as well as layer charge, which is what you are talking about here. You can quantify this using your chemical analyses.

166: are you saying that these elements are in interlayer positions (where they may be accessible to fluids for complexation of organic compounds)? I expect that they are instead in octahedral positions. Again, you need to calculate mineral formulas for the saponites to provide evidence for this assertion.

177: are these assemblages "peculiar" (unusual or strange) or are they particular (specific)?

Fig. 2A: What are the tiny white spots in hematite?

Fig 2C: Why does the CCM have a lath-like shape?

Supplemental Table 11: Why recalculate analyses as Fe₂O₃? Delete the analyses with Fe as FeO and only show Fe as Fe₂O₃.

Fig 2A Caption: you mention chlorite(Ch12) but chlorite2 is not observed in 2A.

RESPONSE TO THE REVIEWERS:

We would like to thank the Referees for carefully reading the manuscript and for their constructive comments. Changes and additions were made accordingly in the revised version of the manuscript (highlighted in blue). Please find below our answers in blue. The line numbers refer to the revised version of the manuscript with changes marked.

Reviewers' comments:

Reviewer #1 (Remarks to the Author):

This is a pioneering report on the abiotic formation of condensed carbonaceous matter in mantle-derived serpentinites of ophiolites. As far as I can recall there are no (or just peripheral) publications on the occurrence of solid abiotic organic matter in mafic or ultramafic rocks on Earth. The findings will certainly have an impact on the view of the deeper - mantle - parts of the ocean floor as constituents of the global carbon cycle, although quantification may be a major challenge for the future. I have only one specific comment; it would be appropriate to include the citation to the following two articles on line 31:

Holm, N.G. and Charlou, J.L. 2001. Initial indications of abiotic formation of hydrocarbons in the Rainbow ultramafic hydrothermal system, Mid-Atlantic Ridge. *Earth and Planetary Science Letters* 191, 1-8.

Charlou, J.L., Donval, J.P., Jean-Baptiste, P., and Holm, N., 2002. Geochemistry of high H₂ and CH₄ vent fluids issuing from ultramafic rocks at the Rainbow hydrothermal field (36°14'N, MAR). *Chemical Geology* 191, 345-359.

That said, I definitely think that this contribution should be published in *Nature Communications*.

We gratefully thank Reviewer#1 for highlighting the novelty of our work and for his/her encouraging comments. We agree with Reviewer#1 that the results presented

in this manuscript may help refining the description of the constituents of the deep carbon cycle. We have added the two proposed references, which now correspond to references 6 and 7.

Reviewer #2 (Remarks to the Author):

This paper documents the occurrence of condensed carbonaceous material (CCM) in oceanic serpentinites, and the authors divide this CCM material into three types on the basis of mineral associations. They use Raman spectra to show that the CCM is not consistent with a biological origin, in contrast to CCM that has previously been identified in seafloor serpentinites. They suggest that minerals catalyze the production of CCM by abiogenic reactions, and infer that this process is important for the global carbon cycle. Serpentinization in general is of great current interest, from microbiologists and geochemists, and their role in cycling of volatiles and other elements is important. I think this paper is timely and worthy of publication, but I see three key questions that need to be answered before it could be ready for publication.

The finding of abiogenic CCM is new and an important part of understanding the origins of reduced carbon in oceanic serpentinites. I am not an expert in organic carbon compounds, but the evidence presented is convincing that the CCM does not have the characteristics of biogenic CCM.

We gratefully thank Reviewer#2 for his/her encouraging comments.

The authors do not, however, address the question of whether this CCM could be derived from original biogenic CCM (see comment no 1 below).

The different occurrences of CCM are linked to different low-T mineral assemblages, and this linkage is convincing. The authors do not, however, provide convincing evidence for some of their proposed mineral replacement reactions (see comment no 2 below). This may be a minor point, because the CCM is clearly associated with these different assemblages. Whether these formed by replacement of other secondary phases or replacement of igneous relics may not affect their arguments

about the formation of CCM. But the authors need to better define the saponite (calculate mineral formulas, see detailed comments below), and do a better job of putting their interpretations into a realistic scenario for uplift and exposure of mantle material on the seafloor (or else do not discuss this at all; see detailed comments). Finally, the authors overstate the importance of serpentinization on the global carbon budget. This is presented as the major significance of this work, but this question has already been addressed in several other studies. See comment no 3 below.

1. What about obduction reactions affecting CCM that originally had a biological origin in the rocks? The authors cite a geologic map for evidence that the rocks are “not affected by metamorphic overprint during orogenic exhumation and obduction.” This is not very convincing. The rocks could easily have been heated to 150-200°C and the minerals not show any effects of metamorphism because they are stable at such conditions.

What happens to biogenic CCM like that observed in seafloor serpentinites if held at 200°C for thousands (tens or hundreds of thousands?) of years? Could this change biogenic CCM into the observed “abiogenic CCM” material? I do not know enough about organic matter diagenesis to know if this is possible, so this may not be significant, but this possibility needs to be addressed. Similar mantle rocks that were serpentinized on the Tethyan seafloor and are located ~80 km west of Casale were subjected to up to prehnite-pumpellyite facies conditions, but show little mineralogical effects of this because there were essentially no fluids present (only minor local effects in fractures/veins).

We acknowledge that this is a critical point. We agree with Reviewer#2 that if these rocks underwent metamorphism up to the prehnite-pumpellyite facies conditions would not necessarily show detectable mineral changes. Unfortunately, little is known on the metamorphism that could have affected the Val Baganza ophiolitic unit despite the cited work of (Marroni et al., 2001). Contrarily to the location mentioned by Reviewer#2 (i.e. in the Internal Ligurides), this ophiolitic unit is part of the External Ligurides. It mainly shows deformations by folding and thrusting and locally very low metamorphic grade (Marroni et al., 2001 – cited in text as ref 23).

The main point is that the carbonaceous material itself *does not* show evidence of late thermal alteration.

The long-time fate of organic molecules depends on several environmental parameters; among them the temperature certainly plays the predominant role then followed by redox changes and interaction with fluids. Thermal degradation of biogenic (and abiotic) carbonaceous material is known to profoundly modify the primary chemical composition and molecular structure of the carbonaceous material. Thermal stress induces a progressive enrichment in carbon and aromatic structures due to the loss of oxygenated groups (as well as S- or N-bearing groups) and of aliphatic units (e.g. Delarue et al., 2016). This process is effective at temperature as low as 50°C. It results in a pronounced change in the Raman spectrum of primary carbonaceous material (see for example, Marshall & Olcott Marshall, 2010). As a result of the progressive aromatization of the organic material, atoms rearrange into graphitic layers. Raman spectra of thermally degraded carbonaceous material display then two broad bands located at 1580-1610 cm⁻¹ and 1340-1360 cm⁻¹, resulting from the in-plane vibration of the aromatic carbon and the out-of-plane vibrations of these aromatic layers, respectively (e.g. Sforza et al., 2014; Delarue et al., 2016; Baludikhay et al., 2018). Although such (hydro)thermal alteration of primary biogenic material has been reported in serpentinites from the Mid-Atlantic ridge (Ménez et al., 2012; Pasini et al., 2013 – refs 14 & 15), we found no evidence for these bands hence of a thermal alteration of the primary carbonaceous material in the Val Baganza ophiolitic samples.

In addition the behavior upon (thermal) alteration of organic compounds also strongly depends on its relationship with minerals. Our group provided notable evidences that organic compounds such as amino acids or organometallic complexes, when intercalated in the interlayers of clay minerals such as smectites, drastically change their thermal behavior (Brigatti et al., 2004; Malferrari et al., 2006; Bernini et al., 2015). Thermal alteration then occurs when starts the thermal decomposition of the octahedral sheet of the clay mineral (at about 600°C). This is much higher than the temperatures at which organic compounds would decompose when not bound to mineral (usually between 200 and 300°C). Overall, we can exclude that the CCM described in this study corresponds to biogenic material that have been thermally degraded during obduction-related metamorphism. Furthermore, the univocal textural relationships that exist between a given CCM type and a given mineral assemblage

suggest that the carbonaceous matter condensed in paragenetic equilibrium with the observed minerals.

We changed the text accordingly lines 45-49 and 122-126.

Lines 45-49: “Our study focuses on a small serpentinitic body (150m-long; Fig. 1B) pertaining to the Val Baganza unit of the External Ligurides²². Mineral paragenesis and nature of the organic matter disseminated in these serpentinites confirm previous studies showing that this unit was not affected by metamorphic overprint during orogenic exhumation and obduction²³”

Lines 122-126: “Accordingly, the CCM spectral signatures reported here (Fig. 5) lack evidence of protein-forming amide groups of biological origin, such as those previously reported in oceanic serpentinites^{14,15}. A CCM genesis after thermal degradation of pristine biogenic material can be excluded based on Raman spectra that do not display the expected broad bands of graphitic carbon that would be produced by this process ($\sim 1340\text{-}1360\text{ cm}^{-1}$ and $1580\text{-}1610\text{ cm}^{-1}$)³¹.”

References cited in this comment:

Baludikay, B. K. *et al.* Raman microspectroscopy, bitumen reflectance and illite crystallinity scale: comparison of different geothermometry methods on fossiliferous Proterozoic sedimentary basins (DR Congo, Mauritania and Australia). *Int. J. Coal Geol.* **191**, 80–94 (2018).

Bernini, F., Castellini, E., Malferrari, D., Borsari, M. & Brigatti, M. F. Stepwise structuring of the adsorbed layer modulates the physico-chemical properties of hybrid materials from phyllosilicates interacting with the μ -oxo Fe+3-phenanthroline complex. *Microporous Mesoporous Mater.* **211**, 19–29 (2015).

Brigatti, M. F., Colonna, S., Malferrari, D. & Medici, L. Characterization of Cu-complexes in smectite with different layer charge location: Chemical, thermal and EXAFS studies. *Geochim. Cosmochim. Acta* **68**, 781–788 (2004).

Delarue, F. *et al.* The Raman-Derived Carbonization Continuum: A Tool to Select the Best Preserved Molecular Structures in Archean Kerogens. *Astrobiology* **16**, ast.2015.1392 (2016).

Malferrari, D., Brigatti, M. F., Laurora, A., Medici, L. & Pini, S. Thermal behavior of Cu(II)-, Cd(II)-, and Hg(II)-exchanged montmorillonite complexed with cysteine. *86*, 365–370 (2006).

Marshall, C. P. & Olcott Marshall, A. The potential of Raman spectroscopy for the analysis of diagenetically transformed carotenoids. *Philos. Trans. R. Soc. A Math. Phys. Eng. Sci.* **368**, 3137–3144 (2010).

Sforna, M. C., van Zuilen, M. A. & Philippot, P. Structural characterization by Raman hyperspectral mapping of organic carbon in the 3.46 billion-year-old Apex chert, Western Australia. *Geochim. Cosmochim. Acta* **124**, 18–33 (2014).

2. Some of the low temperature mineral reactions that are proposed are not convincing.

What is the evidence for replacement reactions among low-T minerals? Specifically, what is the evidence that saponite replaces chlorite etc? What is the evidence that hematite replaces saponite? (ref 41 is misquoted about this on line 130). Such replacement reactions are not seen in oceanic rocks affected by low-T alteration. Typically highly unstable magmatic minerals are replaced by stable low-temperature phases, but low-T secondary minerals can show disequilibrium and sequential overgrowths rather than replacement (e.g., see Alt, J.C., Very Low Grade Hydrothermal Metamorphism of Basic Igneous Rocks. In: Very Low Grade Metamorphism. M. Frey and D. Robinson, eds., Blackwell Scientific, pp. 169-201.

Alt, J.C., Alteration of the Upper Oceanic Crust: Mineralogy, Chemistry, and Processes, in: Elderfield, H. and Davis, E., eds, Hydrogeology of the Oceanic Lithosphere, Cambridge University Press, New York, 456-488. To me it is more likely that saponite replaces relics of magmatic plagioclase, as part of sequential reactions: chlorite replaces plagioclase at >200°C, then saponite (or even mixed-layer saponite-chlorite) replaces remaining plagioclase at lower temperatures. I do not understand how hematite replaces saponite.

The authors should calculate chlorite and saponite mineral formulas. This could help them constrain the conditions of their formation. The presence or absence of chlorite layers in saponite would help constrain temperatures, and the authors could test

whether ferrous or ferric iron (or both) are present (relevant to their arguments about layer charge and organic molecules in interlayer positions).

Following Reviewer #2 suggestion we have calculated chlorite and saponite mineral formulas and reported in Supplementary Tables 5, 7, 9 and 10. These calculations coupled with the highlighted petrographical relationships allow to better constrain the low-temperature alteration sequence and also to better support the reactions that gave rise to the CCM. Overall, the discussed paragenesis are all part of a logical alteration path resulting from increasing oxidizing conditions as occurring when the oceanic mantle is uplifted and exposed to the seafloor.

➤ Mineral formulas and number of atoms contained in chlorite and saponite have been calculated based on the theoretical formulas of clinochlore and trioctahedral smectite, respectively. Data are reported in new **Supplementary Tables 5, 6, 8, 10, and 11**. In order to calculate the formulas we considered the tetrahedral occupancy (8 for both chlorite and saponite), the octahedral occupancy (6 and 12 for saponite and chlorite, respectively), and the number of total negative charges (44 and 56). In order to evaluate the iron oxidation state and its distribution in the tetrahedral and octahedral sheets, we fixed the tetrahedral occupancy at 8.

Chlorite: With the exception of point 1 in Chl1 where only ferrous iron is present, all the chlorites analyzed show both ferric and ferrous iron in octahedral coordination. The total negative charge falls within common values (56) whereas octahedral occupancy is generally lower than 12 (only point 1 in Chl1 is equal to 12) and suggests the presence of octahedral vacancies (common in chlorite). The presence of alkaline elements (Ca, Na and K) could be associated to relicts of an original plagioclase. **Formulas calculated for chlorite are reported in Supplementary Tables 5 and 6.**

Saponite: The first information that we can extract from these calculations is that the measured saponite is highly heterogeneous at a micrometric scale. The presence of ferric iron in octahedral coordination suggests that saponite has an important dioctahedral component (i.e., domains formed by dioctahedral sheets). The presence of dioctahedral layers in a trioctahedral smectite is not uncommon. Saponite, ideally $(M_x^+ \cdot nH_2O)Mg_3^{2+}(Si_{4-x}^{4+}Al_x^{3+})O_{10}(OH)_2$, is a trioctahedral smectite; however

octahedral Fe^{3+} and Fe^{2+} for Mg heterovalent substitutions are also possible, and several saponites with different Mg/ Fe_{total} ratio are described in literature (e.g. Güven, N., 1988). **Formulas calculated for saponite are reported in Supplementary Tables 12. Relevant data are resumed in Supplementary Table 13.**

Here, we hypothesized that saponite derives from the alteration at temperature $<150^{\circ}\text{C}$ of the $\text{Chl2}\pm\text{Srp}\pm\text{Hadr-2}$ assemblage composed of clinocllore \pm lizardite \pm hydroandradite pseudomorphs on plagioclase. We cannot completely exclude that saponite could also have derived from relicts of plagioclase as suggested by Reviewer#2. However, the petrographical relationships between the saponite domains and the $\text{Chl2}\pm\text{Srp}\pm\text{Hadr-2}$ domains as well as the calculations of the saponite formulas support that the saponite mainly derives from the pseudomorphs on plagioclase $\text{Chl2}\pm\text{Srp}\pm\text{Hadr-2}$.

We consider particularly significant the calculation of the formulas for point 8 (Supplementary Table 12 and 13), showing that locally we have an interlayering chlorite/saponite. In fact the high concentration of Mg, in contrast to the relative low concentration of Si and Ca, is in very good agreement with the composition of a mixed layer chlorite/saponite. It was therefore possible to calculate separately the formula of the two minerals (please see Supplementary Tables and 1311). This conclusion is also supported by chemical analyses of saponite in point 9 that nicely agrees with a chlorite not completely transformed into saponite.

This interlayering chlorite/saponite suggests that saponite derives from chlorite present in the ex-plagioclase domains. There are only two chlorites present in these domains (new Fig. 3 and new Supplementary Fig. 1): (1) Chlorite 1 (Chl1), contouring the primary spinels altered in ferritchromite with plagioclase ghost around. Chl1 results from the interaction between the cations released during the alteration of the spinel (Al, Mg mainly) and the surrounding serpentine (See Mellini et al., 2005 – ref 26); (2) Chlorite 2, which is part of the $\text{Chl2}\pm\text{Srp}\pm\text{Hadr-2}$ assemblage that is replacing the plagioclase.

Based on the spatial relationship between the $\text{Chl2}\pm\text{Srp}\pm\text{Hadr-2}$ domains and the saponite domains (new Fig. 3), we genuinely believe that it is the Chlorite 2 with inputs from the hydro-garnet (Hadr-2) (source of Fe^{3+}) and the serpentine (source of Fe^{2+} , e.g. Marcaillou et al., 2011) that is at the origin of the saponite. Indeed, the two

domains are interwoven with a decrease in the chlorite and hydro-garnet content outward in the Chl₂±Srp±Hadr-2 domain. This suggests a progressive alteration of this domain to saponite. Moreover, we can infer from the calculation of the saponite and chlorite formulas that the saponite is highly heterogeneous at a micrometric scale showing different Fe³⁺ and Fe²⁺ contents. This heterogeneity could be linked to the heterogeneous microcrystalline assemblage that gave rise to the saponite. We changed accordingly the text to better support our hypothesis (lines 77-86).

Lines 77-86: *“The third assemblage, made of Fe-rich Mg-saponite and hematite (HemSap; **Fig. 2**) is located only within the plagioclase ghosts, suggesting that it derives directly from plagioclase relicts or from the Chl₂±Srp±Hadr-2 assemblage. Although a replacement of plagioclase relicts cannot be completely excluded, the HemSap domains and the Chl₂±Srp±Hadr-2 domains clearly intertwine (**Fig. 2C**), with a decrease of the chlorite and hydroandradite proportions toward the borders in favour of a progressive inward alteration of the Chl₂±Srp±Hadr-2 domain to saponite. Based on mineral formula calculation (**Supplementary Tables 10, 11**), local interlayering of chlorite within saponite supports local replacement. This overall supports that saponite likely derived from the Chl₂±Srp±Hadr-2 pseudomorphs after plagioclase. Saponite displays high content in Fe (up to ~7%Wt, **Supplementary Table 10**)..”*

In addition, the calculations of the mineral formulas of the saponite support that the saponite of Casale serpentinite displays important unbalanced charges in the octahedral and tetrahedral sheets due to the substitution of Mg(II) by Fe(III) in octahedral sheets and the substitution of tetrahedral Si(IV) by Al(III) and Fe(III). Additionally the Casale saponite shows an excess of octahedral occupancy (>6, points 1-7 in Supplementary Tables 12 and 13). This supports the hypothesis that part of the iron (or of the others cations usually considered in octahedral coordination) is, instead, in the interlayer position. The total negative charge lower than 44 is also in agreement with a possible partial dehydroxylation of the octahedral sheets and the production free molecular hydrogen (Heller-Kallai et al., 1989) that could promote the formation of highly reduced microenvironments and hence

enhance the CCM accumulation process. This charge unbalancing, the presence of interlayer cations and the partial dehydroxylation agree nicely with our hypothesis that this peculiar saponite played an important role in the reduction of abiotic carbon in the serpentinites.

We postulated that hematite grew at the expense of saponite based on the spatial and petrographical relationships between the two phases. We acknowledge that it is not a frequent pattern of alteration but hematite is only found in the ex-plagioclase domains systematically associated with saponite. Conversely, saponite can be found without hematite. Hematite forms rosettes interlocked with relicts of the $\text{Chl}_2 \pm \text{Srp} \pm \text{HAdr}_2$ assemblage and/or saponite and seems to have mainly grown in saponite cracks (new Fig. 2). On this basis, we interpreted that a late aqueous alteration stage of the pseudomorphs of plagioclase occurred and involved fluids more oxidizing than the one giving rise to saponite, as also supported by the thermodynamic calculations of Catalano (2013) (ref 36 in text). In his study, Catalano (2013) shows that during fluid-rock reactions occurring at increasing O_2 fugacity a Water:Rock (W:R) ratio of 1000 will unbalance saponite, giving rise only to nontronite while fluids with a W:R ratio of 1 will give rise to hematite \pm nontronite. If the O_2 content per Rock mass stays under 0.008-0.01, only hematite will be produced from the destabilization of saponite. The presence of dioctahedral sheets interlayered with the trioctahedral saponite layers suggested by the mineral formula calculations (**Supplementary Tables 9 and 10**) tends to support the hypothesis that hematite derives from the destabilization of the saponite but that the oxidizing power of the circulating fluids stayed low. To strengthen this point we changed lines 86-90, and 150-158.

Lines 86-90: *“In the same domains, hematite (**Supplementary Table 12**) forms rosettes interlocked with serpentine, saponite and chlorite. Hematite is always found in association with saponite but the reverse is not true, suggesting that hematite grew at the expense of saponite. Variably altered spinels are commonly preserved within the pseudomorphs (**Fig. 2**).”*

Lines 150-158: *“The saponite enrichment in Fe and its mixed valence can be related to slow alteration at low temperature (<150°C) in the upper most portion of the crust*

during uplift and progressive exposure at the seafloor of mantle-derived rocks to cold and oxidizing seawater-derived fluids at increasing water/rock ratio as the reaction progresses^{41–44}. Similar or close conditions are likely kept during the crystallization of hematite linked to the destabilization of Fe-rich Mg-saponite by oxidizing fluids³⁶. . When saponite is oxidized, nontronite is generally the main by-product at high water/rock ratios (ca. 1000)³⁶. On the contrary, at low water/rock ratios (ca. 1), hematite becomes an additional by-product of the saponite oxidation. It can be the sole if the O₂ dissolved in the fluids is as low as 0.008-0.01 g O₂ per gram of rock³⁶.”

Reference cited in this comment:

Heller-Kallai, L., Milosclavski, I. & Grayevsky, A. Evolution of hydrogen on dehydroxylation of clay minerals. *American Mineralogist* **74**, 818-820 (1989).

Güven, N. (1988) Smectites. In: Hydrous Phyllosilicates (Exclusive of Micas) (S.W Bailey, editor). Reviews in Mineralogy, 19, 497–559. Mineralogical Society of America, Washington, D.C.

Laird, J. (1988) Chlorites: Metamorphic Petrology. In: Hydrous phyllosilicates (exclusive of micas) (S. W. Bailey, editor), Reviews in Mineralogy, 19, 405–453. Mineralogical Society of America, Chantilly, Virginia)

3. The authors imply the significance of serpentinization for the global carbon budget, and say that this is “a potential major controller of the global cycle.”(lines 11 and 25 in the Abstract; lines 182 and 185 in the concluding section). But the amounts and significance of carbon, including organic (reduced) carbon, in seafloor serpentinites has already been documented: see Alt et al Lithos 2013, Chem Geol 2012, and Schwarzenbach et al., GCA 2012, who analyzed serpentinites from the northern Apennines and the modern seafloor for organic carbon contents and isotopic compositions. Alt et al 2013 show that only a few percent of serpentinized mantle material is present at oceanic crustal levels, and the carbon budget in serpentinites is the same as in mafic oceanic crust, so how does this make serpentinization a “major controller”? There are many (hundreds?) of analyses of carbon isotopes for organic carbon in serpentinites, how do these data fit with inorganic origins of CCM vs organic CCM? (See paper by Schwarzenbach et al 2012 GCA for data for Ligurian serpentinites).

We acknowledge that (i) our investigations were performed at a small scale and (ii) processes we highlight in this study only concern late stages of hydrothermal alteration below 200°C. We also acknowledge that, based on the existing measurements, the fraction of organic carbon in abyssal serpentinized peridotites represents generally less than 0.2 wt. % (Früh-Green et al., 2004; Delacour et al., 2008), although Alt et al. (2013) considered that it can reach up to 0.7 wt. % in their estimates of C uptake during low-temperature serpentinization. A proper upscaling projection should also take into account the real extent of the serpentinization layer, that is far to be closely defined at the present-day knowledge. On the other hand the long-term persistency shown by the studied CCM (at least a hundred My) point to a potential importance of this form of carbon sequestration for long term carbon cycles. As also suggested by Milesi et al. (2016) (ref 11 in the text), we believe that this metastable form of organic carbon may constitute a significant fraction for a number of processes: the deep carbon cycle by creating a pool of relatively immobile organic carbon, the extent of H₂ production and the mechanisms of abiotic CH₄ formation in ultramafic environments. We agree, however, that at this stage we are not able to quantify its importance. Text has been adjusted accordingly:

Abstract lines 15-16 (former line 11): “Thermodynamic modelling has recently suggested that condensed carbonaceous matter should be the dominant product of abiotic organic synthesis during serpentinization”

Abstract lines 25-27 (former line 25): “They affect a hybrid mafic-ultramafic paragenesis commonly occurring in the lower oceanic crust, pointing to ubiquity of the highlighted process during serpentinization.”

Concluding section lines 213-215 (former line 182): “Such process could be widespread in mafic/ultramafic-hosted hydrothermal systems at MORs, thus storing effectively organic carbon below the oceanic seafloor in a relatively immobile form.”

Concluding section lines 215-218 (former line 185): “The condensed carbonaceous material can be preserved on the long term like in the Casale samples here described, or serve as carbon sources for deep microbial ecosystems¹⁴, with also the

potential to impact abiotic synthesis pathways including dihydrogen or methane generation¹¹.”.

On the other hand, should be noted that the mooted existence of “many (hundreds?) of analyses” that specifically targeted the organic content in oceanic serpentinites is incorrect. For hard rocks, if we exclude studies on fluid inclusions (as these relate to events above 200°C) and mantle xenoliths, to the best of our knowledge, there are less than 10 published studies that go beyond total carbon content and describe the nature of the organic content in serpentinites (i.e., Sugisaki and Mimura, 1994; Pikovskii et al, 2004; Simoneit et al., 2004; Delacour et al., 2008; Bassez et al., 2009; Manuella et al., 2012; Ménez et al., 2012; Pasini et al., 2013; Pisapia et al., 2018). Half of them are based on bulk rock analysis involving selective extraction of the organic fraction. However, the extraction protocols were not specifically adapted to target carbonaceous phases that can be refractory to methanol dichloromethane and methanol (Delacour et al., 2008; Mateeva et al, 2017) or hexane and dichloromethane (Bassez et al., 2009). We did not elaborate on this aspect in the revised version of the present paper but the lack of a dedicated extraction approach leaves a large uncertainty on the real extent of abiotic condensed carbonaceous matter accumulation in low-temperature.

References cited in this comment:

Alt JC, Schwarzenbach EM, Früh-Green GL, Shanks III WC, Bernasconi SM, Garrido CJ, Crispini L, Gaggero L, Padrón-Navarta JA, Marchesi C (2013) The role of serpentinites in cycling of carbon and sulfur: Seafloor serpentinization and subduction metamorphism. Lithos 178, 40-54

Bassez MP, Takano Y, Ohkouchi N (2009) Organic analysis of peridotite rocks from the Ashadze and Logatchev hydrothermal sites. *Int J Mol Sci* 10:2986-2998

Delacour A, Früh-Green GL, Bernasconi SM, Schaeffer P, Kelley DS (2008) Carbon geochemistry of serpentinites in the Lost City Hydrothermal System (30°N, MAR). *Geochim Cosmochim Acta* 72:3681-3702

Früh-Green GL, Connolly JAD, Kelley DS, Grobety B (2004). Serpentinization of oceanic peridotites: implications for geochemical cycles and biological activity. In: *The Subseafloor Biosphere at Mid-Ocean Ridges*. Wilcock WSD, DeLong EF,

- Kelley DS, Baross JA, and Cary SC, AGU Geophysical Monograph Series, American Geophysical Union, Washington DC, vol 144, p 119–136
- Konn C, Charlou JL, Holm N G, Mousis O (2015) The production of methane, hydrogen, and organic compounds in ultramafic-hosted hydrothermal vents of the Mid-Atlantic Ridge. *Astrobiology* 15:381-399
- Manuella FC, Carbone S, Barreca G (2012) Origin of saponite-rich clays in a fossil serpentinite-hosted hydrothermal system in the crustal basement of the Hyblean Plateau (Sicily, Italy). *Clays Clay Miner* 60:18-31
- Mateeva T., Wolff G.A., Manatschal G., Picazo S., Kuznir N.J., Wheeler J. (2017) Preserved organic matter in a fossil Ocean Continent Transition in the Alps: the example of Totalp, SE Switzerland. *Swiss J. Geosci.* DOI 10.1007/s00015-017-0266-3
- Ménez B, Pasini V, Brunelli D (2012) Life in the hydrated suboceanic mantle. *Nat Geosci* 5:133-137
- Pasini V, Brunelli D, Dumas P, Sandt C, Frederick J, Benzerara K, Ménez B (2013) Low temperature hydrothermal oil and associated biological precursors in serpentinites from Mid- Ocean Ridge. *Lithos* 178:84-95
- Pikovskii YI, Chernova TG, Alekseeva TA, Verkhovskaya ZI (2004) Composition and nature of hydrocarbons in modern serpentinization areas in the ocean. *Geochem Int* 42:971-976
- Pisapia C, Jamme F, Duponchel L, Ménez B (2018) Tracking hidden organic carbon in rocks using chemometrics and hyperspectral imaging. *Sci Rep* 8:2396
- Simoneit BRT, Lein AY, Peresyphkin VI, Osipov GA (2004) Composition and origin of hydrothermal petroleum and associated lipids in the sulfide deposits of the Rainbow field (Mid-Atlantic Ridge at 36°N). *Geochim Cosmochim Acta* 68:2275-2294
- Sugisaki R, Mimura K (1994) Mantle hydrocarbons: abiotic or biotic?. *Geochim Cosmochim Acta* 58 (11):2527-2542

Detailed comments:

31: analogue instead of analogical

We changed the text accordingly (line 33).

38: ...ophiolites, which are lithospheric

We changed the text accordingly (line 41-42).

49: what do you mean by “diffuse” high-temperature assemblage? Do you mean pervasive?

We meant “pervasive” and we changed the text accordingly (line 56).

50: but olivine is stable in hydrothermal systems at $T > 350^{\circ}\text{C}$. Maybe rephrase this sentence to indicate high T breakdown of pyroxene, and then olivine reaction at lower temperatures?

We agree with Reviewer#2 that olivine is stable down to 350°C . The mineral assemblage we report in this paragraph corresponds to typical serpentinization products occurring in the temperature range of $300\text{-}500^{\circ}\text{C}$ (Mével, 2003). The point we want to emphasize here is that compared to the four low temperature assemblages we also report in the Casale serpentinites, the mesh textured lizardite + magnetite and bastite developed at higher temperature. As we focus mainly on the low T assemblages ($T < 200^{\circ}\text{C}$), we let the text as it is. We nonetheless changed the temperature limit to 350°C (line 57).

53: closely is better than strictly

Text has been changed accordingly (line 59).

55: rephrase: ..ferritchromite replacement rims and surrounding plagioclase was replaced by chlorite.

As described by Mellini et al. (2005) (ref 25), we observed chlorite aureoles surrounding the ferritchromite rims resulting from the interaction between the cations (Al^{3+} , Mg^{2+}) released from the spinel and the serpentine (bastite or mesh serpentine after olivine). The chlorite described here is thus not correlated with the alteration of the plagioclase. To clarify our point, we changed the text accordingly (lines 66-69):

Lines 65-68: “*Ferritchromite is commonly surrounded by outward chlorite rims (Chl1), resulting from the interaction between the surrounding serpentine and the cations (Mg^{2+} , Al^{3+}) released during the spinel alteration²⁶ (Fig. 2, Supplementary Fig. 1, and Supplementary Tables 5,6).*”

59: delete “contextually or”

Text has been changed accordingly (line 70).

65: (and elsewhere) pseudomorphs, not pseudomorphoses

Text has been changed accordingly (lines 75, 85, 89, 128, 178, and 438).

70: commonly is better than often

Text has been changed accordingly (line 89).

71: paragenesis, not paragenese

Text has been changed accordingly (line 90).

92: associated with hematite (not to)

Text has been changed accordingly (line 110). We found four other similar typos that were also changed (lines 161, 173, 205 and caption of Fig. 6)

93: what is the evidence for replacement by saponite? Why not saponite replacing relics of plagioclase?

We agree with Reviewer#2 that a contribution from plagioclase relics cannot be totally excluded. However, looking at the spatial distribution of the Fe-rich Mg-saponite and hematite assemblage (HemSap) and Chl2±Srp±Hadr-2 domains (composed of clinocllore ± lizardite ± hydroandradite pseudomorphs on plagioclase), we can observe that the two domains intertwine (Fig. 2). We can also note that the border of the Chl2±Srp±HAdr2 domain shows a decrease of the chlorite and hydrogarnet content toward the border with saponite. From the calculations of the

saponite and chlorite formulas, we can infer that the saponite is highly heterogeneous at a micrometric scale showing different Fe³⁺ and Fe²⁺ content and site occupancies but also interlayering with chlorite (Supplementary Tables 9 and 10). This supports that the saponite arose, in nearly closed system, from the Chl2±Srp±Hadr-2 assemblage, especially from the Chl2 with inputs from the hydro-andradite Hadr-2 (source of Fe³⁺) and serpentine (source of Fe²⁺, ⁴⁶). We changed the text to better support our hypothesis (from lines 79 to 85):

Lines 79-85“Although a replacement of plagioclase relicts cannot be completely excluded, the HemSap domains and the Chl2±Srp±Hadr-2 domains clearly intertwine (**Fig. 2C**), with a decrease of the chlorite and hydroandradite proportions toward the borders in favour of a progressive inward alteration of the Chl2±Srp±Hadr-2 domain to saponite. Based on mineral formula calculation (**Supplementary Tables 10, 11**), local interlayering of chlorite within saponite supports local replacement. This overall supports that saponite likely derived from the Chl2±Srp±Hadr-2 pseudomorphs after plagioclase..”

104: delete “in” at beginning of line; reported here, rather than here reported

Text has been changed accordingly (line 123).

105: lack evidence of protein formingderivation, such as those previously reported....

Text has been changed accordingly (line 123-124).

107: ...amount of CMM...

Text has been changed accordingly (line 127).

111: delete “solely”

Text has been changed accordingly (line 132).

112: ...as the only putative...

Text has been changed accordingly (line 132).

121: ...depth in an oceanic subaxial...

Text has been changed accordingly (line 145).

122: delete "in fact".

Text has been changed accordingly

124: but Fe-saponite contains ferrous iron. It is a trioctahedral phyllosilicate, so it must be ferrous iron. You should calculate mineral formulas so you can examine the effects of varying Fe³⁺/Fe^T.

We agree with Reviewer#2 that iron in Fe-saponite must be ferrous. Calculations of the mineral formulas (now presented in Supplementary Tables 12 and 13) show that most of the iron contained in saponite is ferric leaving Mg the major trioctahedral cation. We agree that we abusively used the term Fe-saponite and changed it to Fe-rich Mg-saponite throughout the text. Please see answer to comment 2 for additional details.

128-130: this comparison is not valid for mantle material uplifted and exposed on the seafloor. These lines describe hydrothermal conditions and processes in typical layered mafic crust and are not applicable to exposed mantle sections. See Alt et al 2013 Lithos for some indication of late, low temperature oxidative effects (seafloor weathering) in the uppermost portions of exposed mantle on the seafloor. See also papers about Ocean Drilling at the Iberian Margin for depth variations in some exposed serpentinized mantle sections.

We agree with Reviewer #2 and now refer to seafloor-exposed mantle sections to discuss late oxidizing alteration by seawater at low temperature. Alt et al 2013 (ref. 44) was also added to the reference list.

Lines 150-155 "The saponite enrichment in Fe and its mixed valence can be related to slow alteration at low temperature (<150°C) in the upper most portion of the crust during uplift and progressive exposure at the seafloor of mantle-derived rocks to cold and oxidizing seawater-derived fluids at increasing water/rock ratio as the reaction

progresses⁴¹⁻⁴⁴. Similar or close conditions are likely kept during the crystallization of hematite linked to the destabilization of Fe-rich Mg-saponite by oxidizing fluids³⁶.”

130: ref 41 never said that hematite grows at the expense of saponite as stated here.

We introduce earlier in text that hematite grows at the expense of saponite (lines 86-89).

Lines 86-89: “In the same domains, hematite (**Supplementary Table 12**) forms rosettes interlocked with serpentine, saponite and chlorite. Hematite is always found in association with saponite but the reverse is not true, suggesting that hematite grew at the expense of saponite.”

161: I can understand that saponite could have ferric iron in some layers, but then it is not strictly saponite but rather a dioctahedral smectite component (nontronite layers). You should calculate mineral formulas for the saponite analyses, then you can perhaps say something about ferric vs ferrous iron contents, as well as layer charge, which is what you are talking about here. You can quantify this using your chemical analyses.

Mineral formulas were calculated and are now presented in Supplementary Tables 9 and 10. Please see our reply to previous points and our answer to comment 2 for additional details.

166: are you saying that these elements are in interlayer positions (where they may be accessible to fluids for complexation of organic compounds)? I expect that they are instead in octahedral positions. Again, you need to calculate mineral formulas for the saponites to provide evidence for this assertion.

We agree with Reviewer#2 that these elements (Cr, Ni, Fe, Mn) usually are in octahedral coordination. However the calculations of mineral formulas highlighted an excess of octahedral occupancy supporting the hypothesis that part of these elements (as well as Fe and Mg) are in interlayer positions. Please see our answer to comment 2 for additional details. Text was implemented according to mineral formula calculations (from lines 189 to 203).

Lines 189-203: “The most abundant CCM accumulations are found in the HemSap domains. We propose they originate from the interplay between the capability of hematite to produce H_2 during its formation^{48,50} and the cation exchange capacity of the saponite structure⁵³. Octahedral Fe^{3+} for Mg^{2+} and tetrahedral Al^{3+} and Fe^{3+} for Si^{4+} heterovalent substitutions in the Fe-rich Mg-saponite silicate layers (**Supplementary Tables 10, 11**) may create charge unbalances in the octahedral and tetrahedral sheets⁵³. Moreover, based on the calculation of the mineral formula (**Supplementary Tables 10, 11**) saponite appears to be highly heterogeneous at the micrometric scale with varying quantity of Fe^{3+} and Fe^{2+} in various coordination numbers. The charge unbalance promotes the exchange of cations in the clay mineral interlayer⁵³. It also provides catalytic acid sites in the tetrahedral layers that may catalyse direct adsorption/intercalation, retention, and polymerization of organic compounds⁵³. Chemical formulas calculated for the Fe-rich Mg-saponite reveal a high octahedral occupancy that nicely agrees with the possible presence of transition metals (Cr, Ni, Fe, Mn, and Mg; **Supplementary Tables 10, 11**) in the saponite interlayers. These cations would further promote the complexation of organic compounds⁵³. Coupled to the H_2 produce during the crystallisation of hematite, this would promote further the CCM reduction.”

177: are these assemblages “peculiar” (unusual or strange) or are they particular (specific)?

We changed peculiar to particular (line 207).

Fig. 2A: What are the tiny white spots in hematite?

To analyze these tiny white spots, we went back to the ESEM after surface cleaning revealing they were an artefact during the acquisition of the first images, potentially charging effect. We replaced the figure 2A (now Fig 3A) by the new image we acquired.

Fig 2C: Why does the CCM have a lath-like shape?

This lath-shape is a polishing artefact during thin section preparation. During polishing, some CCM aggregates ripped from their original position or have been distorted. An example of the original appearance can be seen on the right-bottom corner of Fig. 2C, where organics are molding the hematite.

Supplemental Table 11: Why recalculate analyses as Fe₂O₃? Delete the analyses with Fe as FeO and only show Fe as Fe₂O₃.

We did not recalculate Fe₂O₃ from (FeO)_T. It is a formatting error while compiling the supplementary tables. We thus changed the label in the table with means.

Fig 2A Caption: you mention chlorite (Chl2) but chlorite2 is not observed in 2A.

Chlorite 2 is part of the assemblage Chl₂±Srp±Hadr-2. We modified the figure accordingly (now Fig. 3).

REVIEWERS' COMMENTS:

Reviewer #2 (Remarks to the Author):

The authors have done a thorough job addressing all of my comments on the previous version of the manuscript. I appreciate their discussion of my comments, and the authors have made suitable changes where appropriate. I only have one clarification, and a couple of very minor comments. I think the paper is appropriate for publication, with revisions at the authors' discretion.

Reviewed by Jeff Alt

I think the authors may have misunderstood my comment about C isotope data for organic carbon in serpentinites, and I mis-cited Alt et al 2013, who only give total carbon contents. The other papers I cited do give data for bulk rock total organic carbon (TOC) contents and $\delta^{13}\text{C}_{\text{TOC}}$ for oceanic serpentinites (about one hundred analyses total in Alt et al., 2012 Chem Geol, and Schwarzenbach et al 2013 Geochimica). These data are not for the organic carbon extractions that the authors cite in their rebuttal, but are for reduced carbon remaining in bulk rocks after removal of carbonate carbon by dissolution with HCl. The remaining leached bulk rock containing reduced carbon is combusted and analyzed in an element analyzer and mass spectrometer. As the authors probably know, this is a standard technique to analyze total carbon and total organic (reduced) carbon in bulk rocks. I thought that it would be worthwhile for the authors to consider whether this data can provide any further information about the origins of their CCM, or perhaps the wider significance of the CCM. Are the $\delta^{13}\text{C}$ values of this bulk TOC consistent with their interpretation of inorganic formation of CCM? If so, then the bulk rock carbon data could be used to imply that CCM is a major component of organic carbon in serpentinites and this process may be ubiquitous. Or do the TOC isotope data support a microbial origin for most organic matter in oceanic serpentinites? In this case, the TOC data would suggest that the CCM may be a minor component of the reduced carbon in most oceanic serpentinites. Or perhaps the data are equivocal.

56-57: maybe say $>300^{\circ}\text{C}$, as olivine is stable at 350°C and will not react to mesh texture serpentine at 350°C as stated. This change would agree with your statement about the temperatures that Mevel cites ($300\text{-}500^{\circ}\text{C}$)

161; "...can be the sole product if..."

Answers to the Reviewers' comments.

We gratefully thank Prof. Jeff Alt for careful reading the second version of this manuscript. Responses to the Reviewer#2 comments are provided below in blue and in italics

Reviewer #2 (Remarks to the Author):

The authors have done a thorough job addressing all of my comments on the previous version of the manuscript. I appreciate their discussion of my comments, and the authors have made suitable changes where appropriate. I only have one clarification, and a couple of very minor comments. I think the paper is appropriate for publication, with revisions at the authors' discretion.

Reviewed by Jeff Alt

I think the authors may have misunderstood my comment about C isotope data for organic carbon in serpentinites, and I mis-cited Alt et al 2013, who only give total carbon contents. The other papers I cited do give data for bulk rock total organic carbon (TOC) contents and $\delta^{13}\text{C}_{\text{TOC}}$ for oceanic serpentinites (about one hundred analyses total in Alt et al., 2012 Chem Geol, and Schwarzenbach et al 2013 Geochimica). These data are not for the organic carbon extractions that the authors cite in their rebuttal, but are for reduced carbon remaining in bulk rocks after removal of carbonate carbon by dissolution with HCl. The remaining leached bulk rock containing reduced carbon is combusted and analyzed in an element analyzer and mass spectrometer. As the authors probably know, this is a standard technique to analyze total carbon and total organic (reduced) carbon in bulk rocks. I thought that it would be worthwhile for the authors to consider whether this data can provide any further information about the origins of their CCM, or perhaps the wider significance of the CCM. Are the $\delta^{13}\text{C}$ values of this bulk TOC consistent with their interpretation of inorganic formation of CCM? If so, then the bulk rock carbon data could be used to imply that CCM is a major component of organic carbon in serpentinites and this process may be ubiquitous. Or do the TOC isotope data support a microbial origin for most organic matter in oceanic serpentinites? In this case, the TOC data would suggest that the CCM may be a minor component of the reduced carbon in most oceanic serpentinites. Or perhaps the data are equivocal.

We find the suggestion very interesting. We agree that the $\delta^{13}\text{C}$ signature of the condensed carbonaceous matter could help to upscale to the global C cycle. However, we believe that it would not add to the novelty and meaning of our discovery, i.e. the presence of purely abiotic accumulation processes. The acquisition of such isotopic data is part of an ongoing study on a larger serpentinite collection in order to carefully constrain the biotic vs. abiotic budget to the carbon storage during serpentinization, provided that the condensed carbonaceous matter is not resistant to combustion.. However, given the large temperature fluctuation in the oceanic crustal

system the interrelation between diverse carbon accumulation processes can be delicate to disentangle. A pilot work done by our group showed how biotic and abiotic processes vary through time and space at the micrometric scale (Pasini et al., 2013. Lithos 178, 84-95). This observation by itself means that the published bulk carbon $\delta^{13}\text{C}$ values derive from multiple sources and their meaning should be considered carefully. In the meantime, we believe that the data presented here are unequivocally suggesting the efficiency of mineral-driven carbon accumulation and that the novelty and implications of the present work deserve by themselves publication.

56-57: maybe say $>300^\circ\text{C}$, as olivine is stable at 350°C and will not react to mesh texture serpentine at 350°C as stated. This change would agree with your statement about the temperatures that Mevel cites ($300\text{-}500^\circ\text{C}$)

We changed the text accordingly.

161; "...can be the sole product if..."

We changed the text accordingly.